# Balance of Autonomic Nervous Activity, Exercise, and Sleep Status in Older Adults: A Review of the Literature

**DOI:** 10.3390/ijerph182412896

**Published:** 2021-12-07

**Authors:** Miki Sato, Feni Betriana, Ryuichi Tanioka, Kyoko Osaka, Tetsuya Tanioka, Savina Schoenhofer

**Affiliations:** 1Department of Clinical Nursing, Kochi Medical School, Kochi University, Nankoku 783-8505, Japan; osaka@kochi-u.ac.jp; 2Graduate School of Health Sciences, Tokushima University, Tokushima 770-8509, Japan; fenibetriana@gmail.com (F.B.); taniokaryuichi@gmail.com (R.T.); 3Institute of Biomedical Sciences, Graduate School, Tokushima University, Tokushima 770-8509, Japan; tanioka.tetsuya@tokushima-u.ac.jp; 4Anne Boykin Institute, Florida Atlantic University, Boca Raton, FL 33431, USA; savibus@gmail.com

**Keywords:** autonomic nervous activities, healthy aging, heart rate variability, sleep status, well-being, sedentary lifestyle

## Abstract

While older people are frequently known to experience sleep disturbances, there are also many older people who have a good quality of sleep. However, little is known about the balance of autonomic nervous activity, exercise habits, and sleep status in healthy older adults. This study reviews the literature regarding balance of the autonomic nervous activity, exercise, and sleep in healthy older adults. Relevant articles were searched from electronic databases using the combination of the following keywords: “Autonomic nervous activity”, “sleep status”, “sleep”, “healthy older adults”, “aging”, “heart rate variability (HRV)” and “exercise”. Articles were included if they met inclusion criteria: (1) Published in English, (2) Article types: research and review articles, (3) Main outcome was related to the autonomic nervous activity, lifestyle, sleep, and/or healthy aging, and (4) Fully accessed. From 877 articles that were identified, 16 articles were included for review. Results showed that the autonomic nervous activity changes with increasing age, particularly a constant decline in cardiac vagal modulation due to the significant decrease in the nocturnal parasympathetic activity. In addition, the autonomic nervous activity was also related to sleep status and lifestyle, particularly the capability to exercise. In preparing older people toward a healthy aging, maintaining good sleep quality and exercise is suggested.

## 1. Introduction

“Healthy life expectancy”, which indicates the period during which people can live their daily lives without being bedridden or receiving long-term care, is about 10 years less than the average life expectancy [1]. This situation indicates that during the last decade of their lives, many older people have declined in physical and psychological conditions, implying a poor quality of life. Because older people’s ability to do things will be different from those of younger people, the extended years of life with physical and mental declines may give negative implications for older people and society [2].

One of the common problems influencing the quality of life in older people is poor sleep quality. Older people are known to experience sleep disturbances frequently. It was anticipated that 40–70% of older adults suffered from chronic sleep problems, while up to 50% of the cases were not diagnosed [3]. A study [4] examining sleep quality among 331 older people in institutions found that older people experienced poor sleep quality with pain as the main predictor of this sleep disturbance. Furthermore, it was indicated that older people with dementia had poor sleep quality due to role limitation because of emotional problems, and emotional well-being [4]. Another study [5] identified female gender, depressed mood, and physical illness as the risks for future sleep disturbances. Other predictors of poor sleep quality also included lower physical activity levels, lower economic status, widowhood, loneliness, and stress [5].

Alternatively, there are many older people who are healthy even as they get older, have jobs and hobbies, contribute to society, and maintain their independence. “People who look young” have their own lifestyle and are considered to manage their lives to maintain their youth. Good quality of life among older people is characterized by feeling healthy, being able to manage on their own, spending time doing activities that bring a sense of value and joy, having a close relationship, feeling attached, at peace, and secure [6]. Thus, a good quality of life for older people is associated with a healthy condition physically and psychologically. 

Besides sleep quality, various factors are related to healthy aging. Those include healthy behaviors, such as eating a balanced diet, maintaining a regular activity, quitting tobacco, and improving physical and mental capacity [2]. Another factor that might contribute to healthy aging is autonomic nervous activity which is commonly measured by heart rate variability (HRV). HRV is identified as a marker of healthy aging in which higher resting HRV, older age, and better sleep were found to significantly predict psychological wellbeing [7]. HRV was also found to be associated with sleep duration and quality, which may affect wellbeing. A study conducted among 527 participants showed that those with short sleep duration, low sleep quality, and insomnia showed lower cardiac parasympathetic tone and higher levels of sympathetic tone [8]. 

However, studies that combine autonomic nervous activities, exercise, and sleep status among older people are limited. Therefore, this study aimed to review the literature by combining those three variables in older adults and identify important factors that could be used to guide older people to healthy aging as well as determine opportunities for subsequent research.

## 2. Materials and Methods

### 2.1. Design

A literature review was conducted through the steps of Carnwell and Daly [9], which were (1) identifying the purpose of the literature, (2) exploring the articles using keywords that involve the scope of the literature, (3) organizing the results of the review, and (4) determining the conclusion that will inform further studies. 

### 2.2. Search Strategies

In this review, relevant articles were searched from electronic databases, including PubMed, ScienceDirect, and Google Scholar. The search process used the Boolean operator AND/OR on the combination of the following keywords: “Autonomic nervous activity,” “sleep status,” “sleep,” “healthy older adults,” “aging,” “heart rate variability (HRV)” and “exercise.” Additional relevant articles were also searched by looking at the references of relevant articles. Search strategies are presented in Figure 1. 

### 2.3. Inclusion and Exclusion Criteria

Articles were included for review if they met inclusion criteria: (1) Published in English within the publication date from 2010 to July 2021, (2) Article types include original research and review articles, (3) Main outcome was related to the autonomic nervous activity, sleep status, exercise, and/or healthy aging, and (4) Fully accessed articles, which authors could obtain the copy of the articles. 

This study referred to the older age definition based on the United Nations as the age of 60 years and older [10]. Thus, this review included studies that involved people aged 60 years and older. However, some studies in this review also included younger people for compared groups. For the focus of this review, we described only older subjects and did not elaborate the younger comparison group in the article description (Please see Table 1). In addition, in order to retrieve the most recent study, this study included studies published until 12 July 2021, the date on which the literature review was conducted.

Articles were excluded if the main results were not in line with the purpose of this literature review.

### 2.4. Article Synthesis and Analysis

The relevant articles were screened and appraised by two independent reviewers. The study reports provided a range of information; however, analysis of these study reports did not always indicate significant factors such as presence of comorbidities including cardiovascular disease. Available significant information of each article was extracted into a table, including research subjects, main findings, and limitation of the studies. The main findings were then grouped into three themes. The description of the reviewed articles is presented in Table 1.

## 3. Results

At the initial search using a combination of keywords by Boolean operators “AND” and “OR”, 877 articles were identified. From those numbers, 571 articles were excluded because they did not meet the inclusion criteria. After further screening of full text accessibility, duplicate publication, and further screening of the studies’ findings, 16 articles were included for this review, which were published between 2010 to July 2021. These articles consist of one review article and 15 original research articles with the samples involving older people. The total sample of older people involved in those research articles are 55,868 participants, with female and male ratio is 1.4:1 (women: 32,274, men: 23,594). The smallest number of samples was 12 [25] and the largest number of samples was 43,935 [22]. The minimum age involved in the study was 50 years old [22] and the maximum mean age was 101.9 years [14]. The majority of those study included older people who are free from cardiovascular diseases [14,15,16,17,18,20,24,25].

Findings from these articles were grouped into three sections: (1). HRV conditions and changes in older adults, (2) role of HRV for healthy aging, and (3) factors contributing to healthy aging. 

### 3.1. HRV Condition and Changes in Older Adults 

HRV changes with increasing age. The changes in HRV influence the sleep status among older people. Older, compared to younger adults showed significantly lower vagally mediated HRV. A sleep-specific reduction in parasympathetic modulation suggests that is unique to Non-Rapid Eye Movement (NREM) sleep in older adults [12]. One study found that age and resting HRV were not related, but there was a positive association found between higher HRV two h before sleep and older age [7].

### 3.2. Role of HRV for Healthy Aging 

The HRV plays an important role in healthy aging. HRV was inversely associated with coronary heart disease risk [17]. Higher resting heart rate and lower HRV were associated with worse functional status and higher risk of future functional decline in older adults, independent of cardiovascular disease [18]. Low standard deviation (SD) of the RR series [SD of normal-to-normal interval (SDNN)] values (<19 ms) could be associated with early mortality in centenarians [14].

### 3.3. Factors Contributing to Healthy Aging 

Reviewed articles discussed several factors that contribute to healthy aging. These include maintaining exercise [13,15] and maintaining good sleep quality [7]. 

Maintaining exercise is important for well-being and healthy aging. A study by Queiroz et al. [15] found that strength training which was conducted in intervals of at least 5 days keeps cardiac work elevated for a long period of time among healthy older people. The elevated HR, which implies increasing cardiovascular load might not have clinical implications. However, concern may occur among older people with cardiac problems, therefore it is not recommended for older people with hypertension and/or cardiac problems [15].

Aerobic exercise is important to protect the heart and brain for older people [16]. Exercise training increases parasympathetic tone at rest in both healthy older and young men, which may contribute to the reduction in mortality associated with regular exercise. Sedentary lifestyle and not being physically active were also associated with reduced HRV. Furthermore, physical exercise combined with mental training, for example, mindfulness training, can result in better health and quality of life for older people due to inducing plasticity in the central nervous systems [13]. Hand baths seem to be effective not only for providing comfort and relaxation, but also for activating the sympathetic nerve activity safely and positively [11].

Another important factor contributing to healthy aging is maintaining better sleep quality. Tan et al. [7] found that better sleep quality predicted psychological well-being of older people, while fewer physical and somatic symptoms predicted better sleep quality.

However, good sleep quality is not merely based on the length of sleep duration. It was found that long sleep duration (>8 h) in older people was associated with elevated risk of poor high-frequency values and unfavorable low-frequency values of HRV. Because low-frequency power (LF) values may serve as a marker for biological age, the independent association between long sleep duration and poor LF values suggests that older long sleepers deviate from normal aging and consequently have a higher risk of premature death [23].

## 4. Discussion

The key to a lifestyle that facilitates healthy aging is a balance of regular physical exercise and adequate sleep, which mediates and is mediated by the autonomic nervous system activity. Healthy aging as a relationship between exercise, sleep and autonomic nervous activity is depicted in Figure 2. Considerable detailed knowledge of the dynamic interaction between and among these three factors was brought into clear focus in this review.

In relation to the ability to exercise, sarcopenia is common in older people. It is a geriatric condition that comprises declined skeletal muscle mass, strength and function, leading to the risk of multiple adverse outcomes, including death. Its pathophysiology involves neuroendocrine and inflammatory factors, unfavorable nutritional habits and low physical activity [26]. An improvement in aerobic capacity, easily obtained by regular practice of physical activity, seems necessary to combat age-related decline in biological rhythms. Based on this study and current knowledge, it was recommended that older people should perform at least 30 min per day of continuous physical exercise involving whole body musculature (e.g., gardening, house cleaning, Nordic walking) in order to develop their aerobic capacity [27]. One of the important lifestyle habits that promotes healthy aging is exercise [28]. A high level of aerobic capacity might also improve biological rhythmicity.

The changes in HRV in older adults are frequently reported to be related to exercise and sleep. Regarding the findings of autonomic nervous activities, most of the reviewed articles revealed that the autonomic nervous activity, which is commonly analyzed based on heart rate variability (HRV), changes with increasing age, in which one of the significant changes is a constant decline in cardiac vagal modulation. However, one study [7] found that resting HRV and age were not related, but there was a positive association between higher HRV in 2 h before sleep and older age. In the reviewed articles, HRV was measured mostly during rest [7,14,18,23].

Declines in autonomic nervous regulatory functions can significantly impair the quality of life of elderly patients [29]. Accumulating evidence shows that compared with sleep duration of approximately 7 h, both shorter and longer sleep duration is associated with adverse health outcomes such as cardiovascular disease, diabetes mellitus, and hypertension, and further affects life expectancy and all-cause mortality [30]. Obtaining optimal levels of sleep is associated with better health-related quality of life and reduced premature mortality risk, independent of demographic, behavioral, and biological conditions [31]. 

Low intensity exercise and lack of strength training are associated with poor sleep, and non-practice of household chores is associated with poor sleep time [32]. A study by Sing et al. [33] found that the reduction in the ability of the parasympathetic nerves to regulate heart rate was associated with older people without disease. Moreover, it was reported the combination of sufficient physical activity, a healthy diet, moderate alcohol consumption, and nonsmoking was associated with a 57% lower risk of composite cardiovascular diseases (CVD) and a 67% lower risk of fatal CVD than adherence to none or one of these healthy lifestyle factors. The addition of sufficient sleep to adherence to all four traditional healthy lifestyle factors resulted in a 65% lower risk of composite CVD and an 83% lower risk of fatal CVD [34].

Increased insomnia severity was associated with the decline in gait ability; in particular, participants with worsened insomnia symptoms or sleep problems may increase the potential risk of falling in elderly women [35]. Although some older adults complain of poor night-time sleep or subsequent impairments in daytime functioning, others assume that their difficulties are part of the normal aging process [36]. Self-reported sleep disturbances, including chronic insomnia, are increasingly common with advancing age [37]. Longer sleep and daytime dozing are associated with an increased risk of delirium. Sleep patterns may reflect unmeasured health status [38]. There was an interaction effect between insomnia and female gender on pain and somatic symptoms in adults. Insomnia and poor sleep quality are closely associated with gender differences in pain and somatic symptoms, especially in the adult population [39]. Reductions in duration and quality of sleep and increases in the prevalence of circadian rhythm and sleep disorders with age favor proteolysis, modify the body composition and increase the risk of insulin resistance, all of which have been associated with sarcopenia [26].

Women and the elderly are the most affected with sleeping disorders. Women with a sleep duration of 7 h or more were generally satisfied with their sleep, while women sleeping less than 7 h expressed dissatisfaction. It is pointed out that sleeping problems are strongly associated with mental stress and several socioeconomic factors. Changes in societal and work-related matters that improve women’s socioeconomic and family situations could be expected to be very health-promotive for middle-aged women [40]. Sleep disturbance and insomnia are commonly reported in postmenopausal women [35,41]. Regular participation in physical activity has favorable effects on sleep quality and is a useful component of good sleep hygiene [42]. Thus, it is very important to perform daytime physical activity to ensure proper sleep.

Changes in gonadotropins and sex steroids with aging are also associated with sleep changes in older adults. In men, testosterone levels decrease progressively with aging after 30 years of age. The decreased testosterone with aging may relate to the increased sleep fragmentation in older adults. In women, estradiol levels decrease and follicle stimulating hormone levels increase significantly during the menopause transition and post-menopause. These changes in reproductive hormones have been associated with increased complaints of difficulty falling asleep and staying asleep [43]. 

During the day, it is advised to keep active. Moderate to vigorous intensity physical activity is best, but even light activity during the day can help sleep better at night [44]. It is important to try avoiding a lot of activity just before bedtime. Evening naps were related to earlier wake-up time in the older adults. Bright-light exposure or moderate exercise could be expected to help to promote alertness in the evening [45]. 

Short sleep duration, poor sleep quality, and later bedtimes are all associated with increased food consumption, poor dietary habits, and obesity. Individuals who sleep less are more likely to snack and consume energy-rich foods than those having enough sleep. Insufficient sleep adversely affects physiological and psychological health and impacts certain behaviors (e.g., eating habits, physical activity participation, alcohol consumption). In short, the maintenance of overall health is potentially compromised by not getting adequate sleep [46]. 

There are individual differences in changes with aging [47]. Even though some functions decline with aging, the ability to make decisions and take actions is guided by the life experience and knowledge that has been accumulated as people get older [48,49]. Although physical changes that accompany aging occur in everyone, the decline in cognitive function can be delayed by physical exercise [50]. Lifestyles from a young age also affect our health. Therefore, it is important to take an interest in health from a young age, maintain an appropriate lifestyle, and manage our health by taking regular medical examinations.

From 15 research studies in this review, the female and male ratio is 1.4:1. While biological origins of the female advantage and social factors might be the reason, most of the female advantage was the ability to survive difficult situations during infancy compared to baby boys [51]. The review indicates that studies were conducted on more older women than men. The condition that more older women are involved in studies may highlight that the population of older women is greater than older men, as previously described by several studies [51,52,53].

With the increasing older population year by year, nursing and healthcare facilities received many older adult patients with various health conditions. With the understanding of the factors that contribute to healthy aging, nurses and healthcare professionals are expected to promote exercise, mental health training, and better sleep quality for the better outcome of healthy aging. 

## 5. Conclusions

This study discussed evidence from literature regarding autonomic nervous activity, lifestyle, and sleep status in older people. Most studies report that HRV changes with increasing age and play an important role in maintaining healthy aging. A limitation of this review is that it included only articles published in English, therefore may not cover information published in languages other than English. Another limitation of the review is that it was not possible to match samples on all characteristics that might be relevant, such as presence of specific comorbidities. With the relation between HRV and healthy aging, factors were identified that contribute to the maintenance of healthy aging, including maintaining a regimen of regular exercise, an appropriate sleep pattern, and maintaining good sleep quality. 

## Figures and Tables

**Figure 1 ijerph-18-12896-f001:**
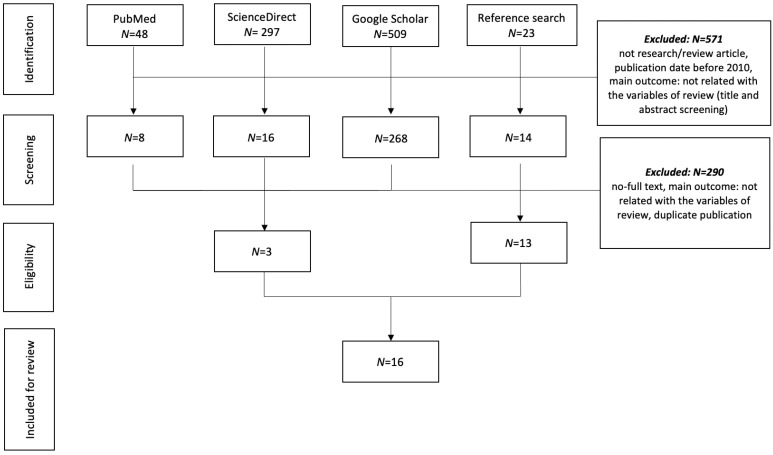
The literature search process.

**Figure 2 ijerph-18-12896-f002:**
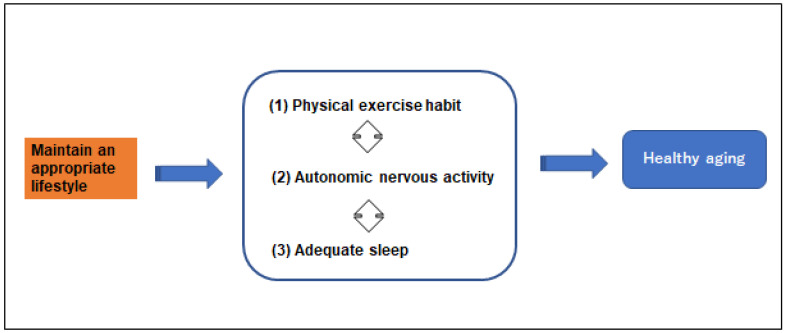
Relationship of maintaining an appropriate lifestyle for healthy aging as physical exercise habit, autonomic nervous activity, and an adequate sleep.

**Table 1 ijerph-18-12896-t001:** The description of the reviewed articles.

Reference Number	Research Subjects	Main Findings	Limitation of the Studies
[7]	▪ Mean age: 67.35 years ▪ *N* = 23 ▪ Characteristics: No known hearing loss, self-reported medical conditions, or medication use that may interfere with autonomic assessment.	▪ HRV was measured during rest using laboratory-based three-lead electrocardiogram (ECG). ▪ Results found that age and resting HRV were not related, but a positive association was identified between higher HRV in the 2 h before sleep and older age. ▪ Higher resting HRV, older age, and better sleep quality significantly predicted psychological well-being, and fewer somatic and physical health symptoms and older age predicted better average sleep quality ratings.	The nature of the study (cross sectional study) makes it difficult to determine the directionality of age, HRV and health associations.
[11]	▪ Mean age: 77.8 years ▪ *N* = 28 ▪ Characteristics: Elderly women without a medical history of conditions such as diabetes.	A hand massage with a warm hand bath improved subjective sleep quality and provided relaxation among older women with disturbance of sleep.	Self-selection bias, in which those who wished to participate were those who were experiencing sleeping problems.
[12]	▪ Mean age: 69.15 years ▪ *N* = 84 ▪ Characteristics: No personal history of neurological, psychological, or other chronic illness.	▪ This study compared the autonomic activity profile between young and older healthy adults during daytime nap and a similar period of wakefulness (quiet wake; QW). ▪ HRV was measured from electrocardiogram data using a modified Lead II Einthoven configuration ▪ Older adults showed significantly lower vagally mediated HRV [measured by root mean square of successive differences between adjacent heart-beat-intervals (RMSSD), high-frequency (HF), low-frequency (LF) power, and total power (TP), HF normalized units (HFnu)] during non-rapid eye movement (NREM) sleep. ▪ No age-related differences were detected during pre-nap rest or QW. ▪ Findings suggest a sleep-specific reduction in parasympathetic modulation that is unique to NREM sleep in older adults.	No standardized diagnostic assessment of psychiatric disorder or chronic medical disease was performed to confirm the self-reported screening surveys.
[13]	▪ Mean age: 64.25 years ▪ *N* = 61 ▪ Characteristics: Free of psychiatric disability and dementia, were not habitual smokers or drinkers, did not take any psychotropic drugs, anti-depressants or cholinesterase inhibitors within the last 12 months.	▪ Mindfulness meditation and physical exercise work in part by different mechanisms, with physical exercise increase physical fitness and integrative body-mind training inducing plasticity in the central nervous system. ▪ Combining physical and mental training may achieve better health and quality of life results for an aging population.	A relatively small sample of subjects with more females.
[14]	▪ Mean age: 84.1 years (Octogenarians group), and 101.9 years (Centenarians group). ▪ *N* = 35 ▪ Characteristics: No acute diseases, heart disease, and not being on cardiac medication.	▪ HRV was measured at rest using electrocardiogram, while RR intervals were recorded using an HR monitor. ▪ HRV indices reflecting parasympathetic outflow all present an age-related reduction could be representative of a natural exhaustion of allostatic systems related to age. ▪ Low standard deviation (SD) of the RR series [SD of normal-to-normal interval (SDNN)] values (<19 ms), which were monitored at rest, could be associated with early mortality in centenarians. ▪ HRV seems to affect exceptional longevity, which could be accounted for by centenarians’ exposome.	Gender heterogeneity in the centenarian group, with 76.5% of the centenarians’ group was women.
[15]	▪ Mean age: 63.3 years ▪ *N* = 16 ▪ Characteristics: No previous diagnosis of cardiovascular or musculoskeletal diseases.	▪ Heart rate and rate pressure product remained higher after the exercise session for up to 4.5 h. ▪ After a single session of strength training, cardiac sympathetic modulation and heart rate remain elevated in elderly subjects, keeping cardiac work elevated for a long time.	The acute effects of a strength training session in sedentary subjects who might respond differently were not evaluated.
[16]	▪ Mean age: 70.9 years for aerobic group, and 70.4 years for stretching group. ▪ *N* = 24 ▪ Characteristics: Free of any cardiovascular or neurological disease.	▪ The aerobic training group increased vagal-mediated HRV parameters which were measured at rest using a Polar Wearlink^®®^ Wind transmitter belt. ▪ The participants in the aerobic training group improved their performance on the Wisconsin card sorting test. ▪ These results highlight the role of aerobic exercise as an important factor to protect heart and brain, and suggest a direct link between exercise, HRV, and cognition in the aged population.	Small sample size, lack of control for other potential confounders such as diet habits and vitamins or Omega 3 supplements, as well as genetic profiles.
[17]	▪ Mean age: 62.5 years ▪ *N* = 2655 ▪ Characteristics: Healthy aging women, absence of self-reported cardiovascular disease (defined as a history of stroke, myocardial infarction, angina, congestive heart failure, peripheral arterial disease, percutaneous coronary angioplasty, or coronary artery bypass graft) at baseline through the ambulatory electrocardiogram assessment.	▪ HRV was measured using a 24-h ambulatory ECG during usual activities. ▪ HRV was found to be inversely associated with coronary heart disease risk in older women.	Findings are not generalizable to males, younger age groups, and non-white minorities.
[18]	▪ Mean age: 75.3 years ▪ *N* = 5042 ▪ Characteristics: Free of poor cognitive function or advanced heart failure, other major arrhythmias and implanted cardiac pacemakers.	▪ HRV was measured at rest using a 12-lead ECG. ▪ Results found that higher resting heart rate and lower HRV were associated with worse functional status and higher risk of future functional decline.	Using a 10-s ECG; so that no significant association of resting heart rate and heart rate variability with functional status could not be shown.
[19]	Articles including older people aged 68 years and older.	▪ This is a review article that discussed effects of exercise on HRV. ▪ Heart rate recovery after exercise is influenced by parasympathetic reactivation and sympathetic recovery to resting levels. ▪ Resistance exercise training appears to have no effect on resting HRV in healthy young adults, while it may improve the parasympathetic modulation in middle-aged adults with autonomic dysfunction. ▪ Acute resistance exercise appears to decrease the parasympathetic activity regardless of age.	Timeline (from which year articles were included) was not mentioned.
[20]	▪ Mean age: 62 years (for CVD group) and 60 years (non-CVD group) ▪ *N* = 2111 ▪ Characteristics: Free of cardiovascular diseases.	▪ A significantly decreased high-frequency (HF) power was found in cardiovascular disease (CVD) group regardless of sleep stages. ▪ For subjects with CVD risks, the changes in multiple HRV metrics were found, especially, decreased HF. ▪ The HRV data were obtained from the Sleep Heart Health Study.	Only conventional HRV metrics were considered in this study.
[21]	▪ Age: 60–72 years old ▪ *N* = 94 ▪ Characteristics: Body mass index between 17 and 28 (self-reported), no present or past psychiatric or neurological disorder (self-reported complaints), no current severe somatic disease.	Older adults had higher autonomic activation at baseline, but their response to sleep deprivation did not significantly differ from the younger adults.	Participants in the non-sleep deprivation condition were not monitored in the lab until the test session but slept at home and arrived in time for the test session at the laboratory.
[22]	▪ Age: 50 to 80 years and older ▪ *N* = 43,935 (Women = 24,434, men = 19,501) ▪ Characteristics: Age 50 and older (no information regarding inclusion and exclusion criteria).	▪ There was no consistent pattern of association between household socioeconomic status and sleep problems. ▪ Poorer self-rated quality of life, greater disability, and feelings of depression and anxiety were all significantly associated with higher prevalence of sleep problems, both in women and men. ▪ There were consistent higher odds of sleep problems associated with a greater degree of disability and with reporting of severe/extreme feeling of depression, in women and men and across populations.	Data on chronic comorbidities and lifestyle behaviors were not collected, preventing any analysis adjusting for these variables. Information about sleep problems was self-reported, which could produce misclassification of true sleep patterns.
[23]	▪ Age: ≥65 years ▪ *N* = 1721 ▪ Characteristics: Individuals who were aged ≥65 and could provide past medical history.	▪ Older adults with long sleep duration have a higher likelihood to have poor high-frequency (HF) and low-frequency (LF) power values of HRV, which were measured at rest from data obtained using a lead I electrocardiogram. ▪ A higher risk of poor LF in older adults with long sleep duration.	The timeframe of recalling sleep duration was 4 weeks, and whether this sleep duration is capable of representing the stable status over a long period remains unknown.
[24]	▪ Mean age: 60 years (men), and 61 years (women). ▪ *N* = 27 (Men: 14, women: 13) ▪ Characteristics: Nonsmokers, had no history of autonomic dysfunction, cardiovascular disease, asthma, or diabetes, and were not prescribed any cardiovascular or antihypertensive medications.	▪ Total sleep duration increased systolic blood pressure in both men and women, but the increases were not different between groups. ▪ In contrast, total sleep duration elicited divergent muscle sympathetic neural activity responses in older men and women. ▪ Muscle sympathetic neural activity burst frequency increased in postmenopausal women, but not older men.	The limitation is total sleep deprivation experimental approach, because it is more common for humans to be exposed to repeated nights of short sleep (i.e., partial sleep restriction).
[25]	▪ Mean age: 64.9 years ▪ *N* = 12 ▪ Characteristics: No history of cardiovascular disease, sleep disorders, drug abuse and medication intake.	▪ During aging, rapid eye movement sleep was associated with a simplification of cardiac control mechanisms that could lead to an impaired ability of the cardiovascular system to react to cardiovascular adverse events. ▪ Aging can be characterized by a reduction in the entropy indices of cardiovascular variability during wake/sleep cycle and that this fall occurs particularly during REM sleep compared to wake and NREM sleep.	No information if there were participants excluded during the experiment.

## Data Availability

The data presented in this study are available in this article.

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
