# Peer review of "Balance of Autonomic Nervous Activity, Exercise, and Sleep Status in Older Adults: A Review of the Literature"

_ijerph, 2021, doi:10.3390/ijerph182412896_

Round 1

Reviewer 1 Report

Thank you for the opportunity to review the manuscript entitled "Balance of autonomic nervous activity, lifestyle, and sleep status in healthy older adults: A review of the literature". 

Please correct the title (of THE literature).

The research is properly done but the studies included must be more detailed (please insert minimum and maximum of a number of participants, minimum and maximum of ages, comorbidities etc... included in the researchers, limitations of the studies etc). Finding some common ideas is not sufficient. 

Reviewer 2 Report

The manuscript has been sufficiently improved to warrant publication in IJERPH.

Reviewer 3 Report

In the submitted work, Miki Sato and colleagues performed a review that discussed the roles of the autonomic nervous system (ANS) and other factors to healthy aging. The topic is relevant and appropriate especially in the context of the aging population in many parts of the world. In particular, the influence of ANS on healthy aging is considered non-trivial. The work reviewed a list of papers with sufficiently large sample size.

I have some concerns:

1) Major: I appreciate the choice of ANS as the main factor for its roles in healthy aging. But healthy aging is a multifactorial problem. How did the authors choose sleep and exercise, among other factors? (Line 20-21, 86-87). Are these factors the ones directly related to ANS functions? For example, exercise and HRV are related, but other factors are related to HRV such as work-related stress and alcohol habit.

2) Minor: perhaps "...operator of AND/OR of the..." (Line 86) shall be "....operator AND/OR on the...". 

3) Minor: Line153/158, repetition on 'Furthermore'.

Round 2

Reviewer 1 Report

Thank you for sending the improved version of the manuscript. I can see that the authors made some important changes. However, some more adds must be done in order to point exactly the purpose of the research and also to meet the findings of the literature review.

  • it is important to point important information about the research: it was about including what kind of population": with/without cardiovascular problems, with/without comorbidities, with/without main disease? It is not clear even from the beginning: inclusion/exclusion criteria must be very clear. Also, the sections Results and Conclusion must point it.
  • one of the most important conclusions of the literature review is related to physical activity. The title and the Abstract give no clue about it. Please refine.
